# Multimodal Tabular Reasoning with Privileged Structured Information

**Jun-Peng Jiang**[1,2,3*]   **Yu Xia**[3]   **Hai-Long Sun**[1,2,3*]   **Shiyin Lu**[3]   **Qing-Guo Chen**[3]
**Weihua Luo**[3]   **Kaifu Zhang**[3]   **De-Chuan Zhan**[1,2]   **Han-Jia Ye**[1,2†]

[1] School of Artificial Intelligence, Nanjing University, China
[2] National Key Laboratory for Novel Software Technology, Nanjing University, China
[3] AI Business, Alibaba Group.
{jiangjp, sunhl, zhandc, yehj}@lamda.nju.edu.cn
{daxiao.xy, running.lsy, qingguo.cqg, weihua.luowh, kaifu.zkf}@alibaba-inc.com

## Abstract

Tabular reasoning requires complex, multi-step information extraction and logical inference, such as aggregation, comparison, or calculation over tabular data. While recent advances have leveraged large language models (LLMs) for reasoning over structured text tables, such high-quality textual representations are often unavailable in real-world settings, where tables typically appear as images. In this paper, we tackle the task of tabular reasoning directly from table images. Our core strategy is to leverage privileged structured information—specifically, the ground-truth structured table data available during training but inaccessible at test time—to enhance multimodal large language models (MLLMs). The key challenges lie in: accurately aligning visual representations with the structured information, particularly mapping the visual evidence to logical steps; and effectively transferring the reasoning skills learned during training to the MLLM for visual inference. To address these, we introduce TURBO (TabUlar Reasoning with Bridged infOrmation), a new framework for multimodal tabular reasoning using privileged information. TURBO benefits from a structure-aware reasoning trace generator based on DeepSeek-R1, which contributes to high-quality modality-bridged information. On this basis, TURBO repeatedly generates and selects advantageous reasoning traces, further enhancing the model's tabular reasoning ability. Experimental results demonstrate that, with limited (9k) data, TURBO achieves state-of-the-art performance ($+7.2\%$ vs. previous SOTA) across multiple datasets.

## 1   Introduction

Tabular understanding focuses on the extraction of information from various forms of tables, such as structured tables and table images [16, 31, 64, 73]. Going a step further, tabular reasoning [91, 77, 89] involves multi-step extraction, integration, and logical inference over tables to derive the final answer, demonstrating its potential in fields such as finance [95], healthcare [63], and scientific research [24].

With the advancement of large language models (LLMs) [4, 49, 8, 18, 75], particularly their increasing reasoning capabilities [48, 55, 21, 74], performing tabular reasoning tasks becomes feasible. Current efforts primarily focus on structured tables, where the input is typically the table's text or structured representations such as Markdown [85, 77, 89]. In contrast, in real-world scenarios, we can only have access to table images or screenshots rather than clean, structured tables [50, 90, 40]. Given the gap

---

[*]Work done during an internship at AI Business, Alibaba Group
[†]Corresponding author, email: yehj@lamda.nju.edu.cn.

between training inputs and deployment conditions, a natural question arises: *can structured tables be leveraged as privileged information [72, 71] to improve the reasoning capabilities of MLLMs?*

Structured tables inherently contain rich information, allowing for fast and precise extraction of the content in each row and column [26, 76]. However, they often struggle to capture the full structural context, particularly in cases where the layout and relationships between rows and columns are complex [40]. On the other hand, while table images provide clear visual cues, current MLLMs still face challenges in effectively extracting and integrating corresponding information, particularly for complex tasks that require multi-step understanding [33, 80].

Beyond the gap between structured tables and images, while recent MLLMs have shown progress in domains such as mathematical reasoning [62, 66], their performance on tabular reasoning tasks remains limited [65, 92]. In our setting, the model is required to extract relevant content from table images, integrate information based on row and column relationships, and then perform logical inference or numerical computation to arrive at the correct answer. This multi-step process places high demands on both perception and reasoning, exposing clear limitations in current models.

Therefore, to enhance multimodal tabular reasoning capability with privileged structured tables, there are two main challenges. Given the complexity of structured tables, not all content is equally relevant to a given question. It is therefore crucial to identify and extract the most relevant information to guide model learning. Furthermore, once relevant content is identified, the next challenge is to effectively leverage it to improve the reasoning ability of MLLMs, enabling them to progressively construct accurate, multi-step reasoning paths.

To address the challenges, we propose TabUlar Reasoning with Bridged infOrmation (TURBO), a framework that integrates structured table information during training to enhance multimodal tabular reasoning. Our approach leverages modality-agnostic reasoning traces as bridged information, facilitating the knowledge transfer from LLMs to MLLMs. Additionally, we employ reinforcement learning techniques to further enhance the model's reasoning capabilities, allowing it to progressively improve its performance on complex tabular reasoning tasks.

Specifically, we construct a new pipeline tailored for tabular reasoning. We start by feeding structured markdown tables and their corresponding QA pairs into DeepSeek-R1 [21], the SOTA reasoning LLM, to generate structure-aware chains of thought to bridge structured tables and images. This process yields high-quality reasoning data in the form of [question, reasoning, answer] triples, which effectively filter out redundant content and emphasize question-relevant reasoning traces. We then conduct supervised fine-tuning (SFT), equipping the MLLM with initial tabular reasoning capabilities. Building on this foundation, by iteratively sampling and selecting the relative advantages in reasoning paths as GRPO method [61], we further enhance the reasoning capability, progressively strengthening the reasoning performance. Experimental results demonstrate that our TURBO achieves significant improvements ($+7.2\%$ over baselines) with limited data, showing high generalization ability and interpretability. Our contributions are threefold:

- We highlight the practical value of using structured tables as privileged information during training to enhance reasoning over table images, as such structured tables are often unavailable at inference time in real-world scenarios.
- We propose a new framework that utilizes privileged structured information to bridge the modality gap and enhance the tabular reasoning capabilities of MLLMs, combining structure-aware reasoning trace generation with iterative optimization.
- Experimental results show that our method achieves substantial performance improvements over datasets with limited training data. Furthermore, the resulting model demonstrates strong robustness and interpretability in its reasoning process.

## 2 Related Work

### 2.1 Tabular Reasoning

Tabular data is of great importance due to its broad applicability in real-world domains, encompassing a variety of learning tasks from standard classification and regression tasks [82, 38, 84, 28] to open-world challenges [29, 6, 30, 27, 39, 83, 5]. Besides, tabular reasoning is a part of tabular understanding, which involves comprehending the information contained within a table and can be broken down into several tasks, such as Table Structure Recognition (TSR) [59, 58], Table Detection

(TD) [19, 35], and Table Question Answering (TQA) [9, 69, 31]. Traditional methods, whether OCR-based [2, 13, 20] or OCR-free [46, 32, 17, 73, 90], have made significant strides in recognizing the structure and content of tables. However, we focus on more challenging tabular reasoning tasks.

Tabular reasoning involves multi-step information extraction and logical inference over tabular data. In the field of tabular reasoning, several recent works have made notable progress. OPENRT [91] is a open-source framework designed for reasoning over tabular data. It enables the reproduction of existing table pre-training models for fair performance comparison and supports rapid development of new models. [15] explores different table representations and directly prompts LLMs and MLLMs, investigating how structural formats influence reasoning capabilities. Chain-of-Table [77] employs in-context learning to iteratively generate operations and update the table, allowing LLMs to construct reasoning chains where each step builds upon the previous output. HIPPO [40] represents tables using both textual and visual modalities, and optimizes MLLMs to learn richer and more comprehensive information with DPO [57]. However, these works primarily focus on reasoning over structured tables and achieve only limited performance. Since structured tables are rarely available in real-world settings, reasoning over table images becomes a more practical choice.

## 2.2 MLLMs for Reasoning

The field of multimodal large language models (MLLMs) has advanced significantly, particularly in integrating visual and textual processing. Modern MLLMs combine visual encoders [56, 68, 87, 25, 67], LLMs, and fusion modules. Models like BLIP-2 [34] and InstructBLIP [14] use a Q-Former to bridge vision encoders and frozen LLMs. MiniGPT-4 [93] combines Q-Former with a linear projector for better alignment between vision and LLMs. LLaVA [37] employs an MLP projector to improve vision-LLM alignment. Alongside these open-source advancements, proprietary models like GPT-4o/o1 [47, 1], Gemini2.5pro [70], and Qwen2.5-VL-Plus/MAX [3] have excelled in benchmarks and real-world applications.

Recent developments in MLLMs have led to significant improvements in reasoning tasks involving both textual and multimodal data [48, 21, 54, 66, 53, 43]. Most current approaches rely on Chain-of-Thought (CoT) techniques [78] to train MLLMs for sequential reasoning. Notable data-driven initiatives include Math-LLaVA [62], which introduced the MathV360K dataset, and MAmmoTH-VL [22], which builds large multimodal CoT datasets at scale. Another research direction focuses on improving vision-text alignment. MAVIS [88] fine-tunes a math-specific vision encoder with a curated set of captions, while Math-PUMA [96] aligns modalities by utilizing the KL divergence in next-token prediction distributions. In our paper, we focus on reasoning over table images, where structural information is not explicitly available and must be inferred through visual cues and multi-step reasoning to reach the correct answer.

## 3 Preliminaries

In this section, we first introduce the notation of multimodal tabular reasoning with privileged structured information task, followed by some preliminary experiments and analyses.

### 3.1 Multimodal Tabular Reasoning with Privileged Structured Information

Tabular reasoning refers to the task of answering questions that require understanding and logical inference over tabular data. This often involves multi-step reasoning, including locating relevant cells, integrating information across rows and columns, and performing operations such as comparison, aggregation, or arithmetic.

In real-world applications, high-quality structured tables $T$, such as markdown tables, are often difficult to obtain. Instead, what is more readily available are table images $V$, such as screenshots or scanned documents. These image-based tables present significant challenges for reasoning, as the model must perceive both the content and the visual layout before performing any logical inference.

To address this, our goal is to enhance multimodal large language models $\mathcal{M}$ to generate a response $X_a$ for the tabular reasoning question $Q$ over table images $V$ by leveraging structured tables $T$ as privileged information during training. While structured tables are *not accessible at inference time*, they offer rich semantics and precise structure that can guide the model *during training*. By

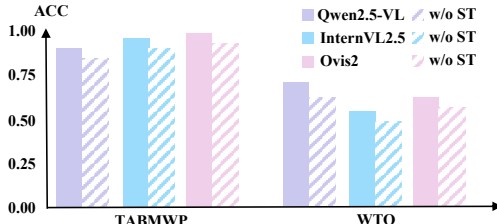

Figure 1: Impact of structured tables (ST). We evaluate the effect of providing structured tables or not as input across Qwen2.5-VL [3], InternVL2.5 [11], and Ovis2 [42] on the TABMWP [41] and WTQ [52] datasets. The results show a consistent performance drop when structured tables are removed, highlighting their importance for effective tabular reasoning.

Figure 2: Challenge of current MLLMs on tabular reasoning. This example illustrates the limitations of current MLLMs in performing multi-step reasoning and numerical computation. The task requires sequential reasoning based on the image, but existing models often lack such capabilities or make errors during intermediate steps, ultimately leading to incorrect answers.

combining visual table inputs with structured supervision, we aim to improve the MLLM's ability to perform accurate and interpretable reasoning over table images. Our goal is for the final answer, $X_{\mathsf{a}}$, to be as consistent as possible with the correct answer $Y_{\mathsf{a}}$.

$$\min_{\mathcal{M}} \sum_{i=1}^{N} \ell(\mathcal{M}(V, Q|T), Y_{\mathsf{a}}) . \tag{1}$$

$\ell$ is the loss function that measures the discrepancy between prediction and ground-truth. "|" indicates that the answer is generated conditioned on the corresponding structured table $T$. Our objective is to enhance the $\mathcal{M}$'s reasoning capability with the privileged structured information. In situations where structured tables is not available in real-world scenarios, we expect $\mathcal{M}$ can still provide accurate answers given only the table image $V$. Therefore, during the inference phase, we measure the performance of $\mathcal{M}$ based on its prediction accuracy given any test image.

## 3.2 Analysis on MLLMs in Tabular Reasoning

To conduct our preliminary study, we select several of the most advanced and widely adopted open-source MLLMs, including Qwen2.5-VL [3], InternVL2.5 [11], and Ovis2 [42], and evaluate them on TABMWP [41] and WikiTableQuestions (WTQ) [52]. We randomly sample 100 examples from each dataset for testing. Since tabular reasoning tasks involve open-ended question answering, it is often challenging to directly judge the correctness of model outputs. We employ Qwen2.5-72B-Instruct [79] as an external evaluator. Specifically, we provide the LLM with the question, ground-truth answer, and the model's prediction. By converting the evaluation into a single-modality setting, this approach reduces ambiguity and improves the reliability of the evaluation, offering a more accurate assessment of MLLMs' reasoning capabilities.

To investigate the role of structured tables in tabular reasoning, we conduct preliminary experiments using MLLMs. Specifically, we compare the model's performance when reasoning is performed with access to structured tables versus relying solely on table images during inference.

**Structured Information Matters in Tabular Reasoning.** Figure 1 presents a comparison between inference results with and without access to structured tables. We observe a significant drop in performance across all models when structured tables are not provided. This highlights the difficulty MLLMs face when relying solely on table images, as they must implicitly recover the table's structure and relevant relationships before performing reasoning. In contrast, structured tables offer clean and explicit content, which greatly facilitates accurate information extraction. These results underscore the importance of leveraging structured representations to guide model reasoning, even if they are only available during training.

**MLLMs Lack Explicit Reasoning Traces.** Both WTQ and TABMWP contain complex, reasoning-intensive questions that require multi-step inference and arithmetic operations to reach the correct answer, as in the case in Figure 2. However, our analysis reveals that current MLLMs often fall

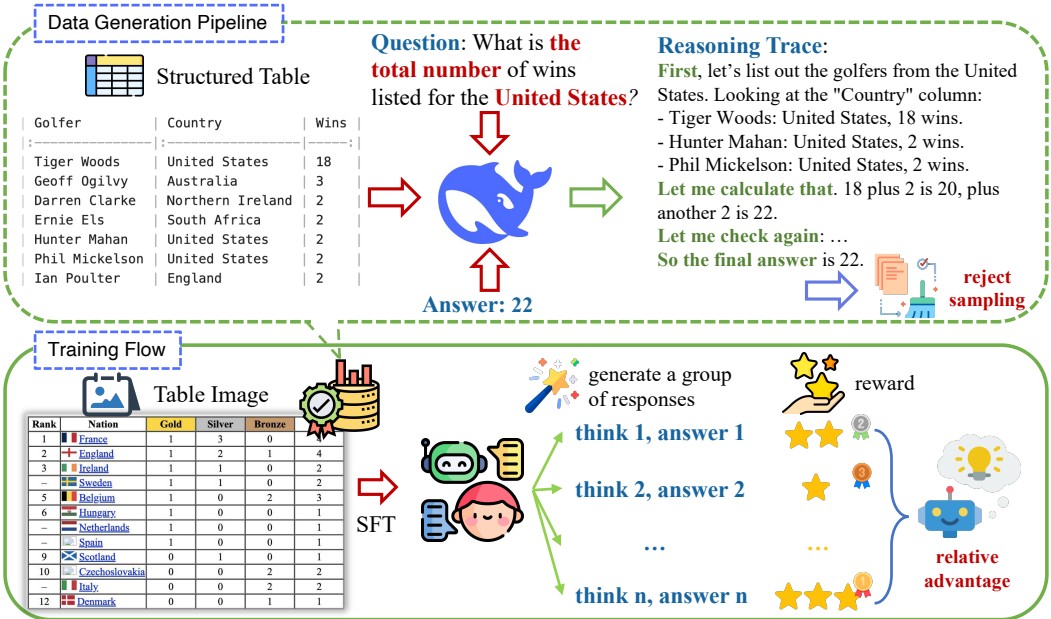

Figure 3: Data generation pipline and training flow of our TURBO framework. We first input the structured table, question, and answer into DeepSeek to generate a reasoning trace, and apply reject sampling to improve data quality. Then, supervised fine-tuning (SFT) is performed on the collected data for privileged information alignment. Finally, we generate a group of answers for each question and compute their relative advantages based on their reward to further reinforce the reasoning ability.

short in these scenarios. Despite having access to table images, these models tend to produce final answers directly, frequently bypassing any clear intermediate reasoning steps. In the absence of explicit prompting for chain-of-thought generation, their outputs rarely exhibit clear reasoning traces. This lack of step-by-step thinking not only undermines interpretability but also frequently results in incorrect predictions. Furthermore, even Qwen shows potential in reasoning traces, there are critical intermediate mistakes, leading to the final wrong answer. These observations highlight the limitations of current MLLMs in tabular reasoning and underscore the need for methods that can explicitly leverage structured reasoning capability and align it with visual representations.

## 4 Multimodal Tabular Reasoning with Bridged Information

In this section, we introduce our TURBO framework (as shown in Figure 3) of multimodal tabular reasoning with privileged structured information through a structure-aware reasoning trace generator. Then we sample and select the advantageous reasoning paths, further enhancing the reasoning ability.

### 4.1 Bridging Structured Tables and Images

A central challenge in multimodal tabular reasoning arises from the intrinsic modality gap between structured tables and table images. These two modalities, though semantically aligned, are processed very differently by MLLMs, leading to differences in their reasoning behavior.

Structured tables, such as those in Markdown or HTML, explicitly encode row-column relationships, enabling accurate and fine-grained access to cell contents [7, 52]. This makes them particularly suitable for tasks that require semantic-level reasoning. However, they often fail to capture complex layout nuances such as merged cells, multi-level headers, or visual hierarchies, which are common in real-world tables [15, 40]. In contrast, table images naturally preserve these structural cues through visual patterns like borders, spacing, and alignment. Yet, due to OCR errors, visual noise, or resolution limits, accurately extracting textual content from table images remains a significant challenge [44]. This inherent modality gap highlights the complementary nature of structured tables and table images—each captures essential but distinct aspects of tabular information.

**Unified Reasoning across Modalities.** However, despite their representational differences, both structured tables and table images ultimately convey the same underlying information relevant to reasoning tasks. From a reasoning perspective, the inference trajectory—*i.e.*, the logical path leading from question to answer—would remain consistent across both modalities. A correct reasoning chain, once established, can guide models to identify the relevant content and apply appropriate reasoning steps, regardless of whether the input is a structured table or its visual counterpart. This highlights that the essence of tabular reasoning lies not in the input format but in the ability to extract and compose the necessary information into a coherent, accurate reasoning process.

Therefore, it is natural to leverage the strong reasoning capabilities of current large language models to extract high-quality reasoning chains for tabular data. These structured chains of thought are modality-independent and can serve as valuable supervision signals for improving multimodal tabular reasoning. To generate structure-aware reasoning traces, we leverage the available structured tables during training. Specifically, for each markdown-formatted table $T$ and its associated question-answer pair $(Q, A)$, we input $(T, Q, A)$ into DeepSeek-R1 [21], a state-of-the-art reasoning LLM. By providing the ground-truth answer $A$ alongside the question and table, we guide the model to produce a coherent and accurate reasoning process $R = \{r_1, r_2, ..., r_n\}$ that logically connects the question to the correct answer.

**Reject Sampling over Reasoning.** This approach produces high-quality supervision triplets $(Q, R, A)$, where the reasoning trace $R$ captures the key logical steps, relevant table elements, and necessary operations to arrive at the correct answer. To further improve data quality, we apply additional reject sampling: overly verbose or contradictory reasoning traces—often caused by inconsistencies between DeepSeek's generated answer and the provided ground-truth answer—are removed. We also employ an auxiliary strong LLM (*e.g.*, Qwen2.5-72B-Instruct) to assess the coherence and correctness of each trace, ensuring that only faithful and focused reasoning chains are retained. After reject sampling, we obtain approximately 9k high-quality examples that serve as reliable supervision for enhancing multimodal tabular reasoning.

In summary, we introduce the bridged information that successfully aligns structured tables with table images through reasoning chains. Our structure-aware reasoning trace generator provides high-quality reasoning traces, facilitating the transfer of reasoning from structured inputs to images.

## 4.2 Enhancing MLLMs with Reasoning Capability

**Preliminary Exploration of Reasoning Capability.** The most straightforward way to enhance MLLMs with reasoning ability is to use supervised fine-tuning (SFT) on our high-quality reasoning data. We construct the following system prompt to guide the model's learning process: `You are a helpful assistant. For each question, first think through your reasoning, then provide an answer. Format your response as: <think>Your reasoning process</think><answer>Your final answer</answer>`. Each training ground truth is formatted as follows: `<think>`$R$`</think><answer>`$A$`</answer>`. Through SFT, this process enables MLLMs to learn the "reasoning-first, answer-later" paradigm. Moreover, since the reasoning traces are modality-invariant, the reasoning ability can be preliminarily transferred from LLMs to MLLMs across modalities.

**Reinforcing Multimodal Reasoning Capability in MLLMs.** Inspired by [21], we adopt Group Relative Policy Optimization (GRPO) as the reinforcement learning algorithm to further enhance the reasoning capability. Unlike SFT, which minimizes token-level prediction errors, GRPO leverages policy gradients derived from reward signals. This enables the model to explore a broader range of possible solutions, encouraging more diverse and complex reasoning behaviors [21, 36].

Specifically, we define questions as $Q$ and the current policy as $\pi_{\theta_{old}}$. For each question $q \in Q$, a group of responses $\{o_1, o_2, ..., o_G\}$ is generated from $\pi_{\theta_{old}}$. A fixed reference policy $\pi_{ref}$ is also used to regularize training. The optimization objective for the updated policy $\pi_\theta$ is defined as:

$$J(\theta) = \mathbb{E}_{q \sim Q, \{o_i\}_{i=1}^G \sim \pi_{\theta_{\text{old}}}}$$
$$\left[ \frac{1}{G} \sum_{i=1}^G \min \left( \frac{\pi_\theta(o_i|q)}{\pi_{\theta_{\text{old}}}(o_i|q)} A_i, \text{clip}\left( \frac{\pi_\theta(o_i|q)}{\pi_{\theta_{\text{old}}}(o_i|q)}, 1-\epsilon, 1+\epsilon \right) A_i \right) - \beta D_{\text{KL}}(\pi_\theta \| \pi_{\text{ref}}) \right] \quad (2)$$

Here, $\epsilon$ is the clipping threshold, and $\beta$ is the weight for the KL divergence penalty. The advantage $A_i$ for each response $o_i$ is computed relative to the group as:

$$A_i = \frac{r_i - \mathrm{mean}\left(\{r_1, r_2, \ldots, r_G\}\right)}{\mathrm{std}\left(\{r_1, r_2, \ldots, r_G\}\right)}. \tag{3}$$

This approach normalizes the reward across the group to obtain a relative advantage signal. The KL divergence term is defined as:

$$D_{\mathrm{KL}}\left(\pi_\theta \| \pi_{\mathrm{ref}}\right) = \frac{\pi_{\mathrm{ref}}\left(o_i \mid q\right)}{\pi_\theta\left(o_i \mid q\right)} - \log\left(\frac{\pi_{\mathrm{ref}}\left(o_i \mid q\right)}{\pi_\theta\left(o_i \mid q\right)}\right) - 1. \tag{4}$$

By leveraging relative advantage, the model is able to perform self-comparison across its own set of generated responses, identifying which responses or reasoning paths are relatively more effective. Meanwhile, the KL divergence term acts as a regularizer, ensuring that the updated policy does not deviate excessively from the reference model in a single optimization step. This constraint helps maintain training stability by preventing abrupt policy shifts, enabling the model to improve progressively in a controlled and stable manner.

Unlike standard PPO [60], GRPO removes the need for a critic network by estimating the relative advantage within each sampled group. This not only simplifies implementation but also significantly reduces computational overhead. In our tabular reasoning setting, the reward function is composed of two parts: format reward and accuracy reward.

The format reward encourages the model to adhere to a structured reasoning format by using `<think></think>` and `<answer></answer>` tags in its response. This design promotes a two-step process where the model first articulates a chain of reasoning based on its understanding of the table, and then explicitly states the final answer derived from that reasoning. By rewarding outputs that follow this format, we guide the model to develop more interpretable and systematic reasoning behavior. The accuracy reward is rule-based and evaluates whether the content in the `<answer></answer>` tags matches the ground truth labels. This component directly incentivizes correctness in the final answer, ensuring that the reasoning process ultimately leads to accurate outcomes.

In summary, reward-guided GRPO efficiently enhances the model's reasoning ability in tabular tasks by selecting superior responses within a group, enabling self-improvement without the overhead of a value function. Through this two-stage learning framework, our model develops a strong tabular reasoning capability, effectively bridging structured and unstructured modalities while enabling robust, interpretable reasoning across table images.

## 5 Experiments

In this section, we first present the experimental setup, including datasets, baselines, and implementation details. We then demonstrate the effectiveness of our approach through comprehensive experiments and ablation studies. Finally, case studies further highlight the interpretability and robustness of our method in complex tabular reasoning scenarios.

### 5.1 Experimental Setup

**Implementation Details.** We conduct all experiments based on the Ovis2 [42]. Our training pipeline consists of two stages: supervised fine-tuning (SFT) and reinforcement learning (RL). During the SFT stage, we train our model using 4×A100 GPUs for 1 hour with a batch size of 128 and a learning rate of 2e-6. This stage allows the model to learn structure-aware reasoning patterns from high-quality [question, reasoning, answer] triples. In the RL stage, we further enhance the model's reasoning ability by applying GRPO with 8×A100 GPUs for 24 hours. We use a smaller learning rate of 5e-7 with the same batch size of 128. For each question, the model generates 16 candidate responses, which are then used to compute relative advantages and guide the optimization process.

**Evaluation Benchmarks.** Following HIPPO [40], we evaluate the tabular reasoning capability of our model on a diverse set of publicly available benchmarks. Specifically, for Tabular Question Answering (TQA) tasks, we use five representative datasets: TABMWP [41], WTQ [52], HiTab [12], and TAT-QA [94]. We export FeTaQA [45] due to its evaluation metric being BLEU [51], which

Table 1: Accuracy comparison across tabular reasoning benchmarks. Our model, TURBO, consistently achieves the best overall performance, demonstrating strong reasoning ability and effective integration of privileged structured table information. Even when compared to the strongest baseline Qwen2.5-VL, TURBO achieves an average improvement of 7.2%.

| Method | Question Answering | | | | Fact Verification | | MMMU | Average |
|--------|--------|------|------|--------|---------|----------|------|---------|
| | TABMWP | WTQ | HiTab | TAT-QA | TabFact | InfoTabs | | |
| InternVL-2.5 [11] | 90.88 | 43.19 | 45.94 | 34.97 | 66.46 | 55.50 | 41.46 | 54.06 |
| Qwen2.5-VL [3] | 92.48 | 65.85 | 67.09 | 70.54 | 83.01 | 77.91 | 42.07 | 71.28 |
| MiniCPM-V-2.6 [81] | 83.68 | 47.97 | 56.53 | 51.55 | 78.48 | 73.03 | 31.10 | 60.33 |
| HIPPO [40] | 87.34 | 55.71 | 63.13 | 61.40 | 82.29 | 75.70 | - | - |
| HIPPO w/o ST | 85.83 | 49.10 | 57.23 | 56.22 | 80.20 | 72.74 | 35.98 | 62.47 |
| Table-LLaVA [92] | 53.20 | 16.62 | 7.87 | 10.49 | 57.62 | 66.78 | 17.68 | 32.89 |
| TabPedia [90] | 10.66 | 23.53 | 6.54 | 13.08 | 35.49 | 2.43 | 2.44 | 13.45 |
| Ovis2 [42] | 92.00 | 58.76 | 68.59 | 47.67 | 80.80 | 74.11 | 48.17 | 67.16 |
| Ovis2-CoT | 92.12 | 60.80 | 66.43 | 48.70 | 81.61 | 72.46 | 50.61 | 67.53 |
| TURBO | **96.75** | **67.80** | **72.15** | **73.21** | **85.81** | **81.89** | **57.32** | **76.42** |

does not focus on the accuracy of question answering. For Table Fact Verification (TFV) tasks, we include TabFact [10] and InfoTabs [23]. To further assess the robustness of our approach, we conduct additional experiments on MMMU [86], evaluating our model on all table-related questions within this challenging multimodal benchmark. This allows us to verify whether the reasoning capabilities learned through our framework generalize to unseen and diverse table images in complex real-world settings. For TQA, we evaluate model performance using Accuracy (ACC), and in TFV, we use the binary classification accuracy for TabFact (true/false outputs) and multi-class accuracy for InfoTabs (entail/contradict/neutral outputs). We employ Qwen2.5-72B-Instruct [79] as an external evaluator. Specifically, we provide the LLM with the question, ground-truth answer, and the model's prediction.

**Comparison Methods.** For comprehensive comparisons, we select leading open-source MLLMs and those in tabular understanding, including Qwen2.5-VL-7B [3], InternVL-2.5-8B [11], MiniCPM-V-2.6-8B [81], Ovis2-8B [42], Table-LLaVA-7B [92], TabPedia-7B [90], and HIPPO-8B [40].

## 5.2 Main Results

In this experiment, we evaluate the table reasoning effectiveness of our TURBO and baseline models on both the TQA, TFV tasks, and MMMU. For the HIPPO method, which leverages both structured tables and table images during training and inference, we report its performance under the original setting with structured tables and table images as input. Additionally, we include HIPPO without structured tables as "HIPPO w/o ST", a variant that takes only table images as input, which better aligns with our setting and serves as a fairer comparison for evaluating purely from table images.

To ensure a fair comparison with HIPPO, we follow the same data sampling strategy by randomly selecting 10k examples in these training datasets as our training set, which do not overlap with our test set. Furthermore, to evaluate the generalization capabilities of TURBO on completely unseen data distributions and formats, we included the challenging MMMU benchmark as an additional test set. We then apply our structure-aware reasoning trace generator to produce reasoning traces. To maintain high data quality, we further conduct reject sampling on incorrect or low-quality generations. As a result, we retain approximately 9k high-quality training samples for SFT and RL for 1 epoch. Results are in Table 1.

Across all datasets, our model TURBO achieves the best overall performance, demonstrating clear and consistent improvements over existing baselines. Specifically, TURBO surpasses the strong tabular understanding baseline HIPPO up to a relative improvement of 21.7% over TQA datasets (in WTQ), and by 8.2% over TFV datasets (in InfoTabs). Notably, compared to the variant HIPPO "w/o ST" (which only uses table images and is thus more comparable to our real-world setting), on average, our method achieves a 26.8% improvement on TQA and 9.8% on TFV, highlighting the effectiveness of our reasoning supervision and privileged structured information.

Furthermore, on the challenging MMMU benchmark, which evaluates table reasoning in real-world multimodal scenarios, TURBO reaches 57.32, outperforming the previous best method Ovis2-CoT

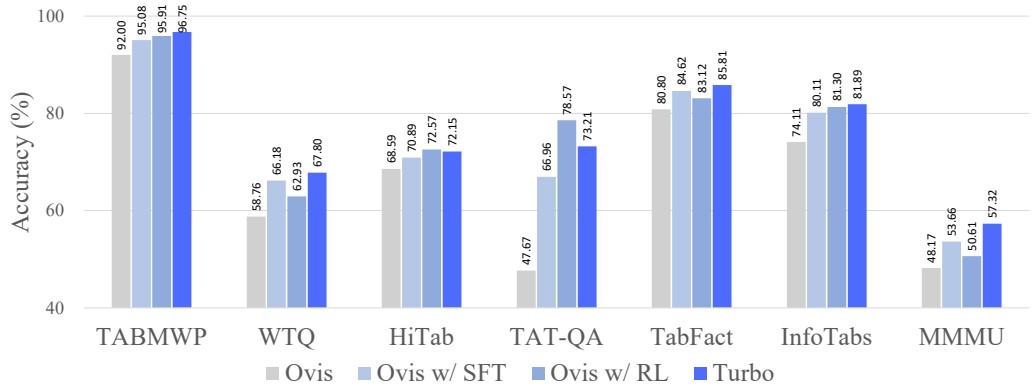

Figure 4: Ablation study of components in TURBO on each dataset. TURBO achieves consistent and significant improvements over the Ovis baseline on all datasets, demonstrating the effectiveness of each introduced component.

| Year | Project A | Project B |
|------|-----------|-----------|
| 0 | -$20,000 | -$24,000 |
| 1 | 13,200 | 14,100 |
| 2 | 8,300 | 9,800 |
| 3 | 3,200 | 7,600 |

**Question**: Maxwell Software, Inc., has the following mutually exclusive projects.Suppose the company uses the NPV rule to rank these two projects. Which project should be chosen if the appropriate discount rate is 15 percent?
A: Project A
B: Project B

**Turbo**: <think>Okay, let's see. The problem is about choosing between Project A and Project B using the NPV rule **with a 15% discount rate**.
**First**, I need to calculate the NPV for each project. **Looking at the table**, the initial investment for Project A is **-$20,000,** and for Project B, it's **-$24,000**. The cash flows for each year. The NPV formula is … discounted by the discount rate (15% or 0.15).
**For Project A**:\nYear 0: -$20,000 (initial investment)\nYear 1: **$13,200 / (1+0.15)^1 = $13,200 / 1.15 ≈ $11,478.26** … NPV_A = -$20,000 + $11,478.26 + $6,279.07 + $2,104.53 ≈ -$138.14
**For Project B**:\nYear 0: -$24,000 (initial investment)\nYear 1: **$14,100 / (1+0.15)^1 = $14,100 / 1.15 ≈ $12,260.87** … NPV_B = -$24,000 + $12,260.87 + $7,401.71 + $4,997.45 ≈ $659.31
**Comparing the NPVs**, Project B has a higher NPV than Project A. Therefore, Project B should be chosen.</think>
<answer>B</answer>

Figure 5: Case study of Turbo. Our method accurately extracts relevant information from the table image (highlighted in blue), formulates the appropriate calculation expressions (in pink), and performs step-by-step reasoning to arrive at the correct final answer (in green). This demonstrates the effectiveness of TURBO in both visual grounding, mathematics, and logical reasoning.

by 13%, and surpassing HIPPO by 59.3% relative improvement, showcasing the robustness and generalization of our model under diverse and complex table formats.

These results validate that our two-stage learning framework not only equips MLLMs with strong reasoning capability, but also highlights its effectiveness in bridging the gap between structured tables and visual table inputs, enabling the model to absorb and utilize privileged structural information.

## 5.3 Further Studies

**Ablation Study.** To better understand the contribution of each component in our two-stage framework, we conducted an ablation study on our baseline model. As shown in Figure 4, we compare the following variants: Baseline: The base model Ovis2 without reasoning supervision. "w/ SFT": Supervised Fine-tuned with our structure-aware reasoning traces. "w/ RL": Optimized with reward-driven reinforcement learning stage. TURBO: The full model trained with both SFT and RL stages.

From the results, we observe consistent improvements across all evaluation tasks. Compared to the baseline, adding SFT alone brings 10.1% relative improvement on average, demonstrating that structure-aware reasoning traces help the model better understand table semantics. The RL stage also introduces 11.8% performance gains, particularly on complex QA datasets like TAT-QA, showing that relative advantage can refine reasoning fidelity and answer consistency. Finally, the full TURBO model achieves the best performance across the board, 13.8% on average. This confirms that both stages in our framework are complementary and jointly crucial for maximizing reasoning ability.

**Case Studies.** To further demonstrate the effectiveness of our approach, we present a case in Figure 5 that showcases the reasoning capability of our TURBO. In this example, the task involves interpreting a table image and answering a complex question that requires multiple-step reasoning, including visual grounding, mathematics, and logical reasoning. More cases and experiment results, such as more ablation studies about the extension of our framework, more experiments on inference time, can be found in Appendix B.

In the baseline model, answers are generated directly without an interpretable reasoning process. After the SFT stage, the model begins to exhibit basic reasoning behaviors but still makes shortcuts, such as incorrect calculations. With RL tuning, TURBO chooses advantageous answers in a group of responses, generates step-by-step reasoning, accurately locates key table elements, and produces interpretable answers, demonstrating clear improvements in both reasoning depth and accuracy.

## 6    Conclusion

In this paper, we propose TURBO, a new multimodal large language model tailored for real-world tabular reasoning tasks, where structured tables are used as privileged information. Our method leverages a two-stage training paradigm that integrates structured reasoning supervision into MLLMs. Specifically, we first introduce a high-quality, structure-aware reasoning trace generation mechanism, enabling the model to internalize a "think-then-answer" paradigm during supervised fine-tuning. We then further refine the reasoning capabilities via reinforcement learning, aligning the model's outputs with more accurate and interpretable reasoning behaviors. Our key contributions to novelty are: 1) a novel application of learning with privileged information; 2) the modality-bridging data heneration pipeline; 3) a synergistic and effective training framework. Extensive experiments show that TURBO consistently achieves state-of-the-art performance among open-source MLLMs. We hope this work inspires future research into multimodal reasoning, especially in developing more systematic and interpretable frameworks for real-world tasks.

## Acknowledgments and Disclosure of Funding

This work is partially supported by National Key R&D Program of China (2024YFE0202800), NSFC (62376118), Collaborative Innovation Center of Novel Software Technology and Industrialization, Postgraduate "AI+" Research and Practical Innovation Project (KYCX25_0328).

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

# A    Training Details

In this section, we will introduce some training details about our experiment, including the datasets we use and the hyperparameters in our experiment.

## A.1    Dataset Details

Following HIPPO [40], we use Table Question Answering (TQA) and Table Fact Verification (TFV) tasks for training and evaluation.

All data statistics are shown in Table 2. we use five representative datasets: TABMWP [41], WTQ [52], HiTab [12], and TAT-QA [94]. We export FeTaQA [45] due to its evaluation metric being BLEU [51], which does not focus on the accuracy of question answering. For Table Fact Verification (TFV) tasks, we include TabFact [10] and InfoTabs [23]. To further assess the robustness of our approach, we conduct additional experiments on MMMU [86], evaluating our model on all table-related questions within this challenging multimodal benchmark.

Table 2: Dataset Statistics.

| Dataset | Train | Test |
|---|---|---|
| Question Answering | | |
| TABMWP | 30,745 | 7,686 |
| WTQ | 17,689 | 4344 |
| HiTab | 8,941 | 1,586 |
| TAT-QA | 5,920 | 772 |
| Fact Verification | | |
| TabFact | 31,321 | 6,845 |
| InfoTabs | 18,383 | 5,400 |
| MMMU | - | 165 |

The construction of the TURBO training dataset utilized TABMWP, WTQ, TAT-QA, TabFact, and InfoTabs. The HiTab dataset is excluded because it involves multi-level tables, which present formatting challenges when converted to Markdown. From each of the chosen datasets, we randomly extract 2,000 instances, resulting in a combined dataset of 10,000 training instances. Each instance consists of a structured table, a question, and the corresponding answer, which are then fed into a structure-aware reasoning tracer generator to generate high-quality reasoning traces. Then, reject sampling is used to further filter the data, resulting in 9,000 training instances in total. This dataset forms the foundation for supervised fine-tuning and subsequent reinforcement learning stages in the TURBO framework.

## A.2    Hyperparameters.

As shown in Table 3, we provide the training hyperparameters for TURBO. Throughout all stages of training, we pre-train for one epoch in 9k generated data, with a batch size of 128.

During the SFT (supervised fine-tuning) stage, we adopt a learning rate of 2e-6 with a cosine decay schedule, a per-GPU batch size of 2, and 16 gradient accumulation steps across 4 A100-80G GPUs. The model is trained with bf16 precision and gradient checkpointing enabled. The maximum text and multimodal input lengths are set to 1500 and 4096 tokens, respectively.

In the RL stage, we fine-tune using a lower learning rate of 5e-7 with a constant schedule. The setup involves 8 A100-80G GPUs with a per-GPU batch size of 1 and the same accumulation steps. Each input prompt generates 16 candidate completions, and we apply a CLIP range of 0.2 to stabilize reward scaling. The sequence lengths are extended to 2000 (text) and 4596 (multimodal). We retain the AdamW optimizer and a warmup ratio of 0.1 across both stages, with no weight decay.

Table 3: Training hyperparameters.

| Config | SFT Stage | RL Stage |
|---|---|---|
| Epoch | 1 | |
| Optimizer | AdamW | |
| Learning rate | 2e-6 | 5e-7 |
| Learning rate scheduler | Cosine | Constant |
| Weight decay | 0.0 | |
| Warmup ratio | 0.1 | |
| Text max length | 1500 | 2000 |
| Multimodal max length | 4096 | 4596 |
| Generation number | - | 16 |
| CLIP range | - | 0.2 |
| Batch size per GPU | 2 | 1 |
| Gradient accmulation steps | 16 | |
| GPU | $4 \times$ A100-80G | $8 \times$ A100-80G |
| Precision | Bf16 | |
| Gradient checkpoint | True | |

## B  More Experiments

In this section, we present additional experimental results to further validate the effectiveness of TURBO. We also include qualitative examples to illustrate the model's behavior and capabilities across various scenarios.

### B.1  Comparison with Reasoning Models

We conducted additional experiments to compare our 8B TURBO model against leading large-scale, reasoning-focused models, even those that operate on structured text inputs, which is an inherently easier task. We compared TURBO with DeepSeek-R1 [21] (671B), a state-of-the-art reasoning LLM, and QvQ [54] (72B), another powerful model designed by Qwen team known for its reasoning capabilities. We also show some text-only results for reference based on our model. The result is shown in Table 4.

Table 4: Comparison with DeepSeek and the text-only results over datasets.

| Method | Question Answering | | | | Fact Verification | | Average |
|---|---|---|---|---|---|---|---|
| | TABMWP | WTQ | HiTab | TAT-QA | TabFact | InfoTabs | |
| DeepSeek-R1 | 98.31 | 85.03 | 82.86 | 86.79 | 91.61 | 79.46 | 87.34 |
| QvQ | 96.38 | 79.54 | 79.03 | 78.26 | 93.25 | 74.91 | 83.56 |
| Baseline-textonly | 94.52 | 62.44 | 75.11 | 66.96 | 82.04 | 72.43 | 75.58 |
| TURBO-textonly | 96.29 | 66.50 | 71.31 | 73.21 | 87.74 | 82.44 | 79.58 |
| TURBO | 96.75 | 67.80 | 72.15 | 73.21 | 85.81 | 81.89 | 79.60 |

This comparison leads to several important conclusions:

**Remarkable Performance Despite Modality Disadvantage:** TURBO operates on raw table images, which requires it to first solve the complex perception problem of parsing the table structure and content before it can even begin to reason. In contrast, models like DeepSeek-R1 are given clean, structured Markdown tables—the very privileged information we use only during training. Despite this significant disadvantage in input modality, our 8B model achieves performance that is highly competitive with a 671B model, and even surpasses DeepSeek-R1 on the InfoTabs dataset. Our model also surpasses QvQ on TABMWP and InfoTabs datasets with smaller model size, which is a larger and stronger reasoning model.

**Exceptional Parameter Efficiency:** Our TURBO framework achieves these strong results with a model that is nearly 85 times smaller than DeepSeek-R1 (8B vs. 671B). This demonstrates that our proposed training methodology is highly effective at distilling complex reasoning abilities into a

much more efficient model. The strength of TURBO is not in model scale, but in its novel technique for bridging the modality gap and transferring reasoning skills.

**Validation of our Core Claim:** This comparison strongly validates our central thesis: by leveraging privileged information during training, we can equip a moderately-sized MLLM with reasoning capabilities on visual tables that approach, and in some cases exceed, those of massive LLMs working with clean, structured data.

Furthermore, we evaluate the generalization capability of our approach by comparing performance under a text-only setting. Specifically, we run both baseline and our TURBO model using structured tables as input. Results show that TURBO, though primarily trained in a multimodal setting, achieves strong performance even when only textual input is provided. The comparable results between TURBO and its text-only variant indicate that our training strategy successfully distills and transfers structured knowledge from tables into the model. This highlights the broader applicability of our approach and its ability to enhance both visual and textual reasoning in a unified framework.

## B.2 Extension of TURBO

As a well-established model, Ovis2 provides a clean and controlled baseline, allowing us to isolate and clearly measure the specific performance gain contributed by our TURBO framework. By applying our methodology to a solid but not top-performing model, we can more transparently demonstrate the effectiveness of our privileged information training and two-stage learning process itself, rather than relying on the sheer power of an underlying SOTA model.

However, to directly address the question about the effectiveness of our framework and to test the generalizability of our approach, we conducted a new experiment applying the full TURBO training pipeline to the stronger Qwen2.5-VL model. The results in Table 5 are highly encouraging and strongly validate the robustness of our framework.

Table 5: Extension of our TURBO framework to Qwen2.5-VL.

|  | TABMWP | WTQ | HiTab | TAT-QA | TabFact | InfoTabs |
|---|---|---|---|---|---|---|
| Qwen2.5-VL | 92.48 | 65.85 | 67.09 | 70.54 | 83.01 | 77.91 |
| Qwen2.5-VL-SFT | 96.02 | 69.77 | 69.73 | 72.59 | 85.92 | 78.17 |
| Qwen2.5-VL-SFT-GRPO | 96.92 | 70.35 | 71.64 | 71.55 | 87.27 | 79.93 |

This experiment leads to two important conclusions: 1) Applying our framework to the already stronger Qwen2.5-VL model yields a consistent and even slightly larger absolute performance gain. This demonstrates that our method of leveraging privileged information provides a fundamental enhancement to tabular reasoning capabilities, regardless of the underlying base model's initial strength. 2) By combining our TURBO framework with a more powerful base model, we achieve better results. This result showcases the scalability of our approach and its potential to push the boundaries of multimodal table reasoning even further when applied to future, more powerful base models.

## B.3 Inference efficiency

Inference efficiency is a critical aspect of any deployable model. We have analyzed the inference speed of TURBO and can provide the following clarification.

Our two-stage training process (SFT + RL) only updates the weights of the model; it does not alter the model's architecture or increase its parameter count. Therefore, the fundamental per-token generation speed of our TURBO model is identical to that of the original Ovis2 baseline. The model itself is not inherently slower.

The observable increase in total inference time for TURBO comes from a deliberate design choice: the model generates a longer output sequence. Instead of producing only a final answer, TURBO generates a full reasoning trace. This naturally takes more time because more tokens need to be generated.

To provide concrete data, we measured the average inference time and output length on the WTQ dataset in Table 6:

Table 6: Inference time in WTQ dataset.

| Model | Avg. Inference Time per Sample (A100-80G) |
| --- | --- |
| Ovis2 (Baseline) | 0.8 seconds |
| TURBO | 2.7 seconds |
| QvQ-72B | 108 seconds |

This data highlights a clear trade-off: TURBO's inference time is longer than its own baseline's because it's performing and articulating a complex reasoning process. This is the cost of achieving significantly higher accuracy and providing a traceable output for error analysis.

Competitive Speed Compared to Other SOTA Models: Crucially, when we compare TURBO's inference speed to that of other powerful, state-of-the-art multimodal models like QvQ, we find that our inference time is not an outlier and remains highly competitive. Despite generating a detailed reasoning trace, TURBO's speed is well within the acceptable limits for models in this performance class. This demonstrates that the efficiency cost for our significant accuracy gains is reasonable and practical.

In summary, the two-stage training does not slow down the model's intrinsic speed. The longer inference time is a direct and predictable consequence of generating more detailed, reasoned outputs. This represents a favorable trade-off for achieving state-of-the-art accuracy, and the resulting speed is comparable to other leading models.

### B.4 More Cases

We present more detailed cases in Figure 6. These cases illustrate the diverse reasoning types and challenges, including multi-step arithmetic, compositional conditions, and fine-grained table comprehension. These cases demonstrate that our TURBO can do well in both visual grounding, mathematics, and logical reasoning.

## C Limitations and Future Works

**Failure cases.** Based on a thorough review of our model's outputs, we have identified two primary categories of failure cases.

OCR Errors in Content Extraction: The first step in visual table reasoning is accurately extracting the textual content. Like most MLLMs, TURBO is susceptible to occasional OCR errors, especially with dense text, small fonts, or image artifacts. These are often subtle but critical. For example, we observed cases where a number like "30001" was misidentified as "3001" or a keyword was misread. Such low-level perception errors inevitably lead to incorrect results in subsequent reasoning steps, as the model is operating on faulty input data.

Errors in Complex Numerical or Logical Reasoning: Even when all table data is extracted perfectly, TURBO can sometimes fail at the final reasoning stage. This is a known challenge for large language models in general. We observed that these errors are most common in tasks requiring multi-step arithmetic calculations or complex logical deductions. For instance, the model might correctly formulate a plan to sum several cells and then subtract a value, but make a mistake during the final calculation. This indicates that while our method significantly enhances reasoning capabilities, it does not completely eliminate the inherent fallibility of LLMs in complex computation.

**Limitations.** While this work focuses on scenarios where structured tables are unavailable at test time—requiring inference purely from table images—there exist many practical situations where both the image and its underlying structured representation can be accessed. Exploring how to effectively incorporate structured tables as additional inputs during inference presents a promising direction for future research.

Another limitation lies in the type of tables considered. Most of our experiments involve relatively clean and regular tables, where each row corresponds to a distinct entity and each column to a specific

**Question**: A candy dispenser put various numbers of orange candies into bags. How many bags had at least 22 orange candies but fewer than 86 orange candies?

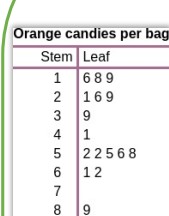

**Orange candies per bag**

| Stem | Leaf |
|------|------|
| 1 | 6 8 9 |
| 2 | 1 6 9 |
| 3 | 9 |
| 4 | 1 |
| 5 | 2 2 5 6 8 |
| 6 | 1 2 |
| 7 | |
| 8 | 9 |

**Turbo**:<think>Okay, let's see. The problem is asking how many bags had **at least 22** orange candies **but fewer than 86**. The data is given in a stem-and-leaf plot.

**First**, I need to understand how stem-and-leaf plots work. The stem represents the tens place, and the leaf is the ones place. So, for example, a stem of 1 with leaves 6, 8, 9 would be 16, 18, 19.

**Looking at the table**, … the stems that fall within this range are 2, 3, 4, 5, 6, and 7. Stem 8 is 89, which is above 86, so that's excluded.

**Now, let's count** the leaves for each of these stems:\n\n- Stem 2: Leaves are 1, 6, 9. That's 21, 26, 29, but 21 should be excluded. So 2 bags …

**Adding these up**: 2 + 1 + 1 + 5 + 2 = 11. So the total number of bags with at least 22 but fewer than 86 candies is 11.</think>
<answer>11</answer>

| Sample No. | Moisture Content | Bulk Density (g/cc) |
|------------|------------------|---------------------|
| 1 | 4.8 | 2.164 |
| 2 | 7.5 | 2.322 |
| 3 | 7.8 | 2.351 |
| 4 | 8.9 | 2.345 |
| 5 | 9.7 | 2.327 |

**Question**: The results of a compaction test on samples of soil that are to be used for an embankment on a highway project are listed below. Determine the optimum moisture content.
A: 10%.
B: 8%.
C: 9%.

**Turbo:** <think>Okay, let's see. The question is asking for the optimum moisture content based on the compaction test results provided in the table. The table lists five samples with their respective moisture content and bulk density.

**First**, I need to understand what \"**optimum moisture content**\" means. In soil compaction, the optimum moisture content is the percentage at which the soil can be compacted to achieve the maximum dry density. …

**Looking at the table**, the samples are numbered 1 to 5. The moisture content increases from **4.8% to 9.7%.** The bulk density values also increase, **starting from 2.164 g/cc and going up to 2.351 g/cc.** …

**The key here is to find** the sample where the bulk density is highest, as that would correspond to the optimum moisture content. Let's check the bulk density values:\n\n- Sample 1: ….\n\nSample 3 has the highest bulk density at 2.351 g/cc, which is the maximum value in the table. …

**Therefore,** the optimum moisture content is 7.8%, which is closest to option B: 8%.</think>
<answer>B</answer>

Figure 6: More cases of Turbo.

attribute. However, real-world tables are often more diverse and complex. For example, datasets like HiTab and MMMU include tables with merged cells, nested headers, and irregular layouts that challenge current MLLMs. A more systematic analysis of how different table structures affect multimodal reasoning performance would be a valuable extension of this work.

**Future work.** Lastly, while our proposed TURBO framework significantly improves the reasoning abilities of open-source MLLMs, a substantial performance gap remains when compared with state-of-the-art proprietary LLMs such as DeepSeek. We hope this paper inspires the community to further explore real-world multimodal reasoning scenarios and to continue narrowing the gap between open- and closed-source models, contributing to more robust and accessible MLLMs.

