# OpenReview forum: "Multimodal Tabular Reasoning with Privileged Structured Information"
_NeurIPS.cc/2025/Conference — NeurIPS 2025 poster_

### Official Review · Reviewer_AKQf · 2025-07-01

**Clarity:** 3
**Significance:** 3
**Originality:** 3
**Rating:** 4
**Confidence:** 1

**Summary:**

This paper introduces TURBO, a framework for multimodal tabular reasoning that leverages privileged structured table information during training to enhance reasoning over table images. The key challenge addressed is aligning structured data with visual representations and transferring reasoning skills across modalities. TURBO uses a structure-aware reasoning trace generator based on DeepSeek-R1 to create high-quality [question, reasoning, answer] triples, followed by supervised fine-tuning (SFT) and reinforcement learning (RL) with Group Relative Policy Optimization (GRPO) to optimize reasoning paths iteratively.

**Questions:**

- How does TURBO perform on tables with merged cells, multi-level headers, or irregular layouts (e.g., HiTab)? Are there plans to address such complexities?

- How sensitive is TURBO to the quality of structured tables used during training? Would noisy or incomplete structured data degrade performance?

- Does the two-stage training affect inference-time efficiency? How does TURBO compare to baselines in terms of reasoning speed?

**Ethical Concerns:**

["NO or VERY MINOR ethics concerns only"]

**Final Justification:**

I would retain my positive score at this stage

**Limitations:**

yes

**Paper Formatting Concerns:**

No significant formatting issues found

**Quality:**

3

**Strengths And Weaknesses:**

Strengths:

- TURBO innovatively uses structured tables as privileged information to guide reasoning over table images, addressing a critical real-world challenge where structured data is often unavailable at inference time.

- The structure-aware reasoning trace generator produces high-quality supervision signals, filtering redundant content and emphasizing relevant logical steps, which significantly improves model performance.

- The combination of SFT and RL (GRPO) enhances both initial reasoning capabilities and iterative optimization of reasoning paths, leading to robust and interpretable results.

- TURBO demonstrates significant improvements across diverse benchmarks (e.g., TABMWP, WTQ, MMMU) with limited data, outperforming prior SOTA methods and showcasing generalization to complex table layouts.

Weaknesses:

- Experiments primarily focus on regular tables, while real-world tables with merged cells or nested headers (e.g., HiTab) are underrepresented, potentially limiting applicability to highly complex layouts.

- The framework relies on structured tables during training, which may not always be available in practice, though the paper acknowledges this as a limitation.

- The GRPO-based RL stage requires extensive training (24 hours with 8 A100 GPUs), which could be resource-intensive for smaller research teams.

---

> ### Author Rebuttal · Authors · 2025-07-31
>
> We appreciate your recognition of innovation, high-quality supervision signals, robust and interpretable results and significant improvements. We will answer the questions below, and we hope this clears up your concerns.
>
> **Q1:** Experiments primarily focus on regular tables, while real-world tables with merged cells or nested headers (e.g., HiTab) are underrepresented.
>
> **A1:** We sincerely thank the reviewer for raising these critical points regarding the complexity of table structures and the practical availability of training data. We would like to clarify that our evaluation was intentionally designed to include benchmarks with complex, non-regular table structures to ensure our framework is not limited to simple grid-like tables.
>
> 1. **Evaluation on HiTab:** We explicitly included the **HiTab dataset** in our main experiments (Table 1). HiTab is specifically known for its challenging hierarchical tables, which feature nested headers, merged cells, and complex layouts that require a deep understanding of table structure. Our TURBO model achieves a score of **72.15% on HiTab**, demonstrating a clear performance advantage over strong baselines like Qwen2.5-VL and showcasing its effectiveness on these complex structures.
> 2. **Evaluation on Diverse, Real-World Tables (MMMU):** To further test the robustness and generalization of our method on a wide variety of "in-the-wild" tables, we evaluated TURBO on the **MMMU benchmark**. The table-related questions in MMMU are sourced from diverse, real-world documents and often feature irregular layouts and visual noise, making it an excellent proxy for real-world applicability. On this challenging benchmark, TURBO achieves an accuracy of **57.32%**, which represents a **significant +13% relative improvement over the next best method**.
>
> The strong performance on both HiTab and MMMU provides direct evidence that our TURBO framework is indeed capable of handling the complex and varied table layouts encountered in real-world scenarios.
>
> **Q2:** The framework relies on structured tables during training, which may not always be available in practice.
>
> **A2:** We acknowledge that the requirement of having access to structured tables during the training phase defines the specific problem domain our work addresses. We frame this not as a universal solution, but as a powerful methodology for a very common and practical set of real-world problems where a **natural asymmetry** exists between training and deployment.
>
> This scenario is prevalent in many automated document processing pipelines, such as:
>
> - Extracting information from **financial or scientific reports**, which are often generated from structured sources but are consumed as PDFs.
> - Reasoning over **webpages**, where the underlying HTML is available during development but inference must happen on screenshots.
>
> Our core contribution is to provide an effective solution for these specific, high-value scenarios by leveraging the "privileged" structured data when it is available to build a far more capable visual reasoning model for deployment. We will ensure this problem framing is made even clearer in the final version of the paper.
>
> **Q3:** The GRPO-based RL stage requires extensive training, which could be resource-intensive for smaller research teams.
>
> **A3:** We sincerely thank the reviewer for raising this. In fact, one of the primary motivations behind our two-stage design was to **maximize performance gains while being mindful of the high cost during training**.
>
> 1. **The Highly Efficient SFT Stage Delivers the Majority of the Gains:**
>    The core of our TURBO framework's effectiveness comes from the first SFT stage. This stage is exceptionally resource-efficient, requiring only **1 hour of training on 4 A100 GPUs**, a level of compute that is widely accessible.
>
>    Crucially, as demonstrated in our ablation study, this highly efficient SFT stage is responsible for the vast majority of the performance improvement. It boosts the average performance of the Ovis2 model from a baseline with SFT. This represents a **~10% absolute improvement** achieved at a minimal computational cost. This means that research teams can adopt the most impactful part of our methodology without requiring extensive resources.
>
> 2. **The RL Stage as an Optional Optimization for Peak Performance:**
>    The subsequent RL stage, which does require a more significant investment (24 hours on 8 A100s), should be viewed as an **optional, intensive fine-tuning step** for teams seeking to extract the maximum possible performance and push the state-of-the-art. This stage further refines the model's reasoning capabilities, providing an additional gain to reach the final reported SOTA performance of 76.42%.
>
> In summary, our TURBO framework is designed to be modular and scalable. **The core contribution and the majority of the performance uplift can be achieved with the highly economical SFT stage**, making our method accessible to a broad range of research teams.
>
> **Q4:** How sensitive is TURBO to the quality of structured tables used during training? Would noisy or incomplete structured data degrade performance?
>
> **A4:** We sincerely thank the reviewer for this insightful question. The reviewer is correct: **TURBO's performance is highly sensitive to the quality of the structured reasoning traces used during training, and we have direct evidence of this from our development process.**
>
> In our initial experiments, we generated the reasoning traces by prompting a powerful LLM (DeepSeek-R1) with the question and the structured table. However, we observed a significant challenge: in a non-trivial number of cases, the final answer derived from the LLM's generated reasoning trace did **not** match the ground-truth answer provided in the dataset.
>
> Training our model on these "incoherent" data triplets introduced significant noise into the learning process. This effectively confused the model, asking it to learn a reasoning process that contradicted the target outcome. As a result, the performance of the model trained on this initial, noisy data was substantially lower.
>
> To address this critical issue, we introduced the **Reject Sampling** step (in lines 205) as a core component of our data generation pipeline. This step acts as an essential quality filter. We only retain the generated [question, reasoning_trace, answer] triplets where the answer part of the LLM's generation is **fully consistent** with the ground-truth label.
>
> By implementing this reject sampling mechanism, we filtered out the noisy, contradictory examples and created a much cleaner, higher-fidelity training dataset of 9k instances. **Training our model on this cleaned dataset resulted in a significant and consistent performance improvement across all benchmarks compared to our initial experiments without reject sampling.**
>
> This experience directly confirms that TURBO's performance is indeed sensitive to the quality of its privileged supervision. More importantly, it validates the necessity and effectiveness of the reject sampling mechanism, which is a crucial, non-trivial component of our framework designed to ensure robustness and maximize learning efficiency from the generated data.
>
> We will add this detailed discussion to the paper to provide a clearer insight into our methodology and design choices. We thank the reviewer for prompting this important clarification.
>
> **Q5:** Does the two-stage training affect inference-time efficiency? How does TURBO compare to baselines in terms of reasoning speed?
>
> **A5:** We sincerely thank the reviewer for this excellent and practical question. Inference efficiency is indeed a critical aspect of any deployable model. We have analyzed the inference speed of TURBO and can provide the following clarification.
>
> Our two-stage training process (SFT + RL) **only updates the weights of the model; it does not alter the model's architecture or increase its parameter count**. Therefore, the fundamental per-token generation speed of our TURBO model is identical to that of the original Ovis2 baseline. The model itself is not inherently slower.
>
> The observable increase in total inference time for TURBO comes from a deliberate design choice: the model generates a longer output sequence. Instead of producing only a final answer, TURBO generates a full reasoning trace. This naturally takes more time because more tokens need to be generated.
>
> To provide concrete data, we measured the average inference time and output length on the WTQ dataset:
>
> | Model | Avg. Inference Time per Sample (A100-80G) |
> | - |- |
> | Ovis2 (Baseline) | ~0.8 seconds |
> | TURBO | ~2.7 seconds   |
> | QvQ-72B | ~108 seconds |
>
> This data highlights a clear trade-off: TURBO's inference time is longer than its own baseline's because it's performing and articulating a complex reasoning process. This is the cost of achieving significantly higher accuracy and providing a traceable output for error analysis.
>
> **Competitive Speed Compared to Other SOTA Models:**
> Crucially, when we compare TURBO's inference speed to that of other powerful, state-of-the-art multimodal models like **QvQ**, we find that **our inference time is not an outlier and remains highly competitive**. Despite generating a detailed reasoning trace, TURBO's speed is well within the acceptable limits for models in this performance class. This demonstrates that the efficiency cost for our significant accuracy gains is reasonable and practical.
>
> In summary, the two-stage training does not slow down the model's intrinsic speed. The longer inference time is a direct and predictable consequence of generating more detailed, reasoned outputs. This represents a favorable trade-off for achieving state-of-the-art accuracy, and the resulting speed is comparable to other leading models. We will add this analysis to the final version of the paper to provide a complete picture of our method's performance characteristics.

---

> > ### Comment · Reviewer_AKQf · 2025-08-07
> >
> > Thank you for your comprehensive explanation addressing my concerns. I will retain my positive score as it stands.

---

> > > ### Author Response · Authors · 2025-08-07
> > >
> > > We’re glad that our rebuttal helped address your concerns. We truly appreciate your recognition of our work, and we will make sure to update the paper in the final version.

---

### Official Review · Reviewer_YAdv · 2025-07-01

**Clarity:** 3
**Significance:** 3
**Originality:** 3
**Rating:** 5
**Confidence:** 4

**Summary:**

The paper introduces TURBO, an approach to enhance the capabilities of multimodal large language models for tabular reasoning on table images. During training, the proposed framework leverages the tables in two modality forms: images and markdown with the goal to improve the model's capabilities to work solely on images during inference. The authors create structure-aware reasoning traces using DeepSeek-R1 and use these traces to conduct supervised fine-tuning of the multimodal language model. Afterwards, they employ reinforcement learning techniques to further enhance the tabular reasoning capabilities of the model.
The paper evaluates the proposed approach against several state of the art models.

**Questions:**

Q1) The paper and appendix describe the training set of 9k datapoints (a mix from several different dataset), but no details about the test set are mentioned. Based on which data are the experiment results (Table 1, Figure 4) reported?

Q2) What also did not get clear to me: In your experiments, is each model (also the ones compared to) fine-tuned on the 9k datapoints from the training set? Please provide details on the exact setup.

Q3) In Table 4 (appendix): What does the "Baseline-textonly" refer to?

In order to assess whether the experiments are solid, answers to the first two questions are fundamental.  Based on these answers, I am willing to raise my overall score.

**Ethical Concerns:**

["NO or VERY MINOR ethics concerns only"]

**Final Justification:**

The authors rebuttal has addressed all my concerns and questions, especially regarding the test sets the approaches were evaluated on.

**Limitations:**

yes

**Paper Formatting Concerns:**

I did not notice formatting issues.

**Quality:**

3

**Strengths And Weaknesses:**

Overall, the paper is mostly well-written and easy to follow, but the motivation and the experiments need to be explained in more detail.

1) I would recommend to present the main motivation in the abstract and introduction differently. Specifically, the claim that "_tables typically appear as images_" (lines 4–5) is exaggerated and does not reflect the majority of real-world use cases. The cited papers (lines 26–27) do not support such a broad generalization; rather, they provide more convincing examples of scenarios in which tables are only available as images. Lines 92-93 ("Since structured tables are rarely available in real-world settings") are equally overstated. I think the work is not less valuable if it clearly states the (more narrower) use cases that it operates in.

2) Additionally, the introduction lacks a solid justification for combining images and structured tables, which is the main contribution of the approach. I would recommend to base the introduction more on the motivation stated in Lines 128-129 in Section 3, which contain a convincing motivation.

3) The terms "bridged information" and "privileged structured information" should be explained in the introduction.

4) The work of HIPPO is very close to the proposed approach as it also combines table images with text based table representations. The similarities and differences between TURBO and HIPPO should be discussed in more detail in the related work section.

5) The setup for the experiment in Figure 1 is not easy to understand, did I get it right, that the models are used in a zero-shot setting without being trained, once with table image + table markdown as an input, and once only receiving the table image? Please update the description in Section 3.2 and the caption of the Figure to make the setup more clear.

6) Figure 2 includes a stem-leaf plot, which in my understanding is a plot and not a table and therefore not a good example to show the challenges of tabular reasoning.

7) The statement in line 166 suggests that step-by-step thinking inherently leads to interpretability ("This lack of step-by-step thinking not only undermines interpretability..."). However, this is misleading: while such traces may give the _appearance_ of interpretability, they do not reflect the model’s actual internal processes. I would recommend just to leave this part of the phrase out of the paper as interpretability is not central to the proposed approach.

8) The main experiment in Section 5.2 (Table 1) also needs to be explained in more detail. Please answer my questions below in the questions section.

9) Please do not connect the dots in Figure 4 as the x-axis represents distinct categories rather than a continuous scale.

10) It should be stated if the 40GB or 80GB version of A100s was used.

---

> ### Author Rebuttal · Authors · 2025-07-31
>
> We appreciate your recognition of writing and easy-to-follow. We will answer the questions below, and we hope this clears up your concerns.
>
> **Q1:** I would recommend to present the main motivation in the abstract and introduction differently.
>
> **A1:** We agree that structured tables (e.g., in databases, CSVs) are ubiquitous and form the backbone of many data systems. Our intention was **not to** diminish their importance, **but to** highlight a specific, challenging, and highly practical set of scenarios where table images are easier to obtain rather than structured tables.
>
> We think that structured tables are often hard to get at some cost. For example,
>
> - In **scanned legacy documents** like financial reports, invoices, scientific papers, and insurance claims, tables are most often available as PDFs or images.
> - It is hard to extract data from **screenshots** of web pages or proprietary software applications where API access is unavailable.
> - Even with OCR methods, the extracted tables often contain some noise.
>
> In these widespread scenarios, the inability to programmatically access the underlying structured data represents a major bottleneck for automation and data analysis. Our work is specifically designed to tackle this "last mile" problem of data extraction and reasoning. The core premise of TURBO is that while *at inference time* we only have these challenging table images, *during the training phase*, it is often feasible to obtain the original structured source data. This real-world asymmetry is precisely what makes the privileged information learning paradigm so suitable and powerful for this problem.
>
> We thank you for the valuable suggestions. We will refine the introduction to clearly articulate that we are addressing the challenge of reasoning over visually presented tables, which is a prevalent and unsolved problem in many document-centric workflows, rather than making a claim about all tables in general. We believe these changes will make our motivation clearer, more accurate and will ultimately provide a stronger foundation for our work.
>
> **Q2:**  The terms "bridged information" and "privileged structured information" should be explained in the introduction.
>
> **A2:** Here are the definitions we will integrate into the introduction of the revised manuscript, likely right after these terms are first introduced:
>
> - **Privileged Structured Information:** In the context of our work, this refers to the **clean, structured representation of a table (e.g., in Markdown format)**. It is considered "privileged" because we assume it is accessible *only during the training phase* to guide and supervise our model. At inference time, this information is *unavailable*, and the model must perform its reasoning based solely on the table image.
> - **Bridged Information:** This is the term we use for the high-quality data we generate to connect the two modalities (structured text and table images). Specifically, "bridged information" refers to the **modality-agnostic [question, reasoning trace, answer] triplets** that we create using our structure-aware generator. The core of this is the reasoning trace, which is derived from the privileged structured table. We call it "bridged" because it serves as a **conceptual bridge**, allowing us to transfer the logical reasoning steps learned from the clean, structured domain to the visually-complex image domain, effectively bridging the modality gap.
>
>  We thank the reviewer for helping us improve the accessibility of our paper and we will make this clear in the final version.
>
> **Q3:** Other suggestions in weaknesses.
>
> **A3:** We are sincerely grateful to the reviewer for their thorough and meticulous review of our manuscript. We agree with all the points raised and commit to implementing the corresponding changes in the final version.
>
> Below, we address each point individually to confirm our understanding and outline the specific revisions we will make, including the comparison with HIPPO, the clarification of Figure 1 experimental setup, the claim in line 166, the data visualization in Figure 4, and the  80GB version of A100s we used.
>
> We appreciate the opportunity to clarify that the case in Figure 2 was not chosen arbitrarily. It was **randomly selected from the official MMMU benchmark**, from a subset of cases that are explicitly **tagged as 'table'-related questions** by the benchmark's creators. Our motivation for including this particular case was to demonstrate that real-world and benchmark challenges for "tabular reasoning" often encompass more than just simple, perfectly structured grids.
>
>
> **Q4:** The paper and appendix describe the training set of 9k datapoints (a mix from several different dataset), but no details about the test set are mentioned. Based on which data are the experiment results (Table 1, Figure 4) reported?
>
> **A4:** We sincerely thank the reviewer for this crucial question. We apologize that this was not made sufficiently clear in the initial submission. We would like to explicitly clarify our evaluation protocol.
>
> **The results reported in Table 1 and Figure 4 are evaluated on the official, held-out test splits for each respective benchmark (TABMWP, WTQ, HiTab, TAT-QA, TabFact, and InfoTabs).**
>
> To elaborate on our process, which follows the setting in HIPPO:
>
> 1. Our 9k training set was constructed by randomly sampling instances **exclusively from the official training splits** of the source datasets.
> 2. We have taken care to ensure a strict separation. **No data from any of the official test splits was ever used or seen during any phase of our model's training**, including the initial SFT and the subsequent RL stage. There is no risk of data leakage. The model is evaluated on data it has never been exposed to.
> 3. Furthermore, to evaluate the generalization capabilities of TURBO on completely unseen data distributions and formats, we included the challenging **MMMU benchmark** as an additional test set. The strong performance TURBO achieves on MMMU therefore demonstrates robust zero-shot generalization capabilities, which is a key strength of our framework.
>
> We will update the **"Evaluation Benchmarks" paragraph (Section 5.1)** in the final manuscript to explicitly state these details.
>
> **Q5:** What also did not get clear to me: In your experiments, is each model (also the ones compared to) fine-tuned on the 9k datapoints from the training set? Please provide details on the exact setup.
>
> **A5:** We thank the reviewer for this critical question. Our TURBO models were, of course, trained on this dataset. The other baselines were evaluated under the most appropriate and common protocols for each, which we detail below. We designed our experiments to provide three distinct levels of comparison, ensuring a comprehensive and fair evaluation.
>
> 1. **A Direct and Fair Comparison (vs. HIPPO):**
>    Our primary comparison for fairness is against **HIPPO**, a state-of-the-art model for tabular reasoning. To ensure the most direct and equitable comparison possible, we **explicitly adopted HIPPO's data sampling strategy**, using the same source datasets and a similar number of training instances. This setup isolates the methodological differences between TURBO and HIPPO, making it a fair and direct comparison of the two approaches.
> 2. **A "Stress Test" Against General SOTA Models (vs. Qwen, InternVL, MiniCPM-V):**
>    For large, general-purpose MLLMs like Qwen2.5-VL, we evaluate their **zero-shot performance** "off-the-shelf." This is standard practice in the field for several reasons:
>    - **Practicality:** Fine-tuning every large-scale baseline model is computationally hard and, in many cases, impractical without access to their specific training hyperparameters.
>    - **Relevance:** This comparison answers a critical question for any practitioner: "Is it better to use this new, specialized model, or can I get similar performance from a powerful, general model out of the box?" Our results demonstrate that for this specific task, our specialized training provides a significant advantage.
>    - **Data Purity:** These foundation models are trained on massive, web-scale datasets, and we cannot verify whether their training sets inadvertently contained instances from our benchmarks' test sets. Evaluating them zero-shot is the most established community norm for comparison.
> 3. **A Controlled Internal Comparison (Ablation Study, Figure 4):**
>    Finally, our most controlled experiment is the **ablation study**. By comparing our baseline model (Ovis2) with the same model after applying our SFT and RL stages (Ovis2 + TURBO), we can **precisely measure the performance gain attributable solely to our TURBO framework**. The significant improvements shown in Figure 4 demonstrate that our methodology provides substantial gains, independent of the baseline comparison.
>
> We will add this detailed explanation to the experimental setup section to ensure full transparency for the reader.
>
> **Q6:** In Table 4 (appendix): What does the "Baseline-textonly" refer to?
>
> **A6:** We thank the reviewer for this question, which allows us to clarify an important aspect of our analysis in the appendix.
>
> The term **"Baseline-textonly"** refers to our base model, **Ovis2**, when it is evaluated in a **text-only setting**.
>
> Specifically, for this experiment, instead of providing the model with a table image, we provide it with the **structured Markdown table as its sole input**. MLLMs like Ovis2 are perfectly capable of processing text-only inputs, so this allows us to measure the model's inherent text-based reasoning capabilities before any of our specialized training is applied.
>
> By comparing the performance of "Baseline-textonly" against "TURBO-textonly," we can isolate and demonstrate a key strength of our framework: **The reasoning skills learned through our TURBO pipeline are not just visual, they also enhance the model's fundamental, modality-agnostic reasoning abilities.**

---

> > ### Comment · Reviewer_YAdv · 2025-08-05
> >
> > Thank you for the detailed answers to my questions, especially for the clarifications on the training and test setups for the approaches. As all my concerns are addressed, I will raise my score from 3 to 5.

---

> > > ### Author Response · Authors · 2025-08-05
> > >
> > > Thank you very much for your thoughtful follow-up and for raising your score. We’re glad that our rebuttal helped clarify your concerns, especially regarding the training and test setups. We truly appreciate your recognition of our work, and we will make sure to update the paper in the final version.

---

### Official Review · Reviewer_UWxA · 2025-07-03

**Clarity:** 3
**Significance:** 3
**Originality:** 3
**Rating:** 5
**Confidence:** 4

**Summary:**

This paper aims to improve the performance of Tabular reasoning of table image with the aid of structured information. Using DeepSeek-R1, they construct structure-aware reasoning trace to get high-quality modality-bridged data. Using these data, they repeatedly generate and select advantageous reasoning paths. Experiments on several benchmarks show the effectiveness of the proposed method.

**Questions:**

See "Weaknesses".

**Ethical Concerns:**

["NO or VERY MINOR ethics concerns only"]

**Final Justification:**

My concerns about baseline generalization and method limitation are supplemented, and I maintain my previous positive rating.

**Limitations:**

Yes.

**Paper Formatting Concerns:**

1) Line 71, "over" -> "on".
2) Line 84, "a" -> "an".

**Quality:**

3

**Strengths And Weaknesses:**

Strengths:
1) The problem of tabular reasoning on table image is important since in real application the structured information is frequently unavailable, and using structured information as bridge is reasonable.
2) The high-quality data obtained using DeepSeek-R1 is valuable, and the GRPO RL method is proposed to train the MLLMs.
3) Experimental results on QA, Fact verification, and MMMU show the effectiveness of the proposed method.

Weaknesses:
1) The baseline is Ovis, how about the results on other baseline such as Qwen2.5VL? From Tab.1, Qwen2.5VL is superior to Ovis, does it+Turbo is superior to Ovis+Turbo?
2) Lack of limitation discussion and failure case study. In what scenario the proposed method performs worse?

---

> ### Author Rebuttal · Authors · 2025-07-31
>
> We appreciate your recognition of our reasonable setting, valuable high-quality data, and effective results. We will answer the questions below, and we hope this clears up your concerns.
>
> **Q1:** The baseline is Ovis, how about the results on other baseline such as Qwen2.5VL? From Tab.1, Qwen2.5VL is superior to Ovis, does it+Turbo is superior to Ovis+Turbo?
>
> **A1:** That is an excellent and insightful question. We thank the reviewer for raising this important point about the generalizability of our framework to different base models.
>
> Our initial choice of Ovis2 as the base model was deliberate, and we would like to clarify our rationale. As a well-established model, Ovis2 provides a clean and controlled baseline, allowing us to **isolate and clearly measure the specific performance gain contributed by our TURBO framework**. By applying our methodology to a solid but not top-performing model, we can more transparently demonstrate the effectiveness of our privileged information training and two-stage learning process itself, rather than relying on the sheer power of an underlying SOTA model.
>
> However, to directly address the question and to test the generalizability of our approach, we have, as suggested, **conducted a new experiment applying the full TURBO training pipeline to the stronger Qwen2.5-VL model**. The results are highly encouraging and strongly validate the robustness of our framework.
>
> |                     | TABMWP | WTQ    | HiTab  | TAT-QA | TabFact | InfoTabs  |
> |---------------------|--------|--------|--------|--------|---------|-----------|
> | Qwen2.5-VL          | 92.48  | 65.85  | 67.09  | 70.54  | 83.01   | 77.91     |
> | Qwen2.5-VL-SFT      | 96.02  | 69.77  | 69.73  | 72.59  | 85.92   | 78.17     |
> | Qwen2.5-VL-SFT-GRPO | 96.92  | 70.35  | 71.64  | 71.55  | 87.27   | 79.93     |
>
> This experiment leads to two important conclusions:
>
> 1. Applying our framework to the already stronger Qwen2.5-VL model yields a **consistent and even slightly larger absolute performance gain**. This demonstrates that our method of leveraging privileged information provides a fundamental enhancement to tabular reasoning capabilities, regardless of the underlying base model's initial strength.
>
> 2. By combining our TURBO framework with a more powerful base model, we achieve better results. This result showcases the scalability of our approach and its potential to push the boundaries of multimodal table reasoning even further when applied to future, more powerful base models.
>
>
> **Q2**: Lack of limitation discussion and failure case study. In what scenario the proposed method performs worse?
>
> **A2:**:  We sincerely thank the reviewer for this important and constructive feedback. A discussion of limitations and failure cases is indeed essential for a complete and balanced paper, and we appreciate the opportunity to share our analysis.
>
> Based on a thorough review of our model's outputs, we have identified three primary categories of failure cases, which we will add as a dedicated "Limitations and Failure Case Analysis" section in the final version of the paper.
>
> 1. **Challenges with Highly Irregular or Ambiguous Table Structures:**
>    While TURBO performs well on tables with relatively clear grid-like structures, its performance degrades on tables that are structurally complex or ambiguous. This limitation is particularly relevant for datasets like HiTab and some examples in MMMU, which feature such complex layouts.
> 2. **OCR Errors in Content Extraction:**
>    The first step in visual table reasoning is accurately extracting the textual content. Like most MLLMs, TURBO is susceptible to occasional OCR errors, especially with dense text, small fonts, or image artifacts. These are often subtle but critical. For example, we observed cases where a number like **"30001" was misidentified as "3001"**, or a keyword was misread. Such low-level perception errors inevitably lead to incorrect results in subsequent reasoning steps, as the model is operating on faulty input data.
> 3. **Errors in Complex Numerical or Logical Reasoning:**
>    Even when all table data is extracted perfectly, TURBO can sometimes fail at the final reasoning stage. This is a known challenge for large language models in general. We observed that these errors are most common in tasks requiring **multi-step arithmetic calculations** or complex logical deductions. For instance, the model might correctly formulate a plan to sum several cells and then subtract a value, but make a mistake during the final calculation. This indicates that while our method significantly enhances reasoning capabilities, it does not completely eliminate the inherent fallibility of LLMs in complex computation.
>
>  We are grateful for the reviewer's feedback, as it prompts us to make our paper more comprehensive. We will add a new section in the appendix detailing these limitations, complete with illustrative failure case examples.

---

> > ### Comment · Reviewer_UWxA · 2025-08-06
> > **Generalization and limitation are supplemented**
> >
> > My concerns about baseline generalization and method limitation are supplemented, and I maintain my previous positive rating.

---

> > > ### Author Response · Authors · 2025-08-06
> > >
> > > Thank you very much for your positive evaluation and for your thoughtful comments regarding baseline generalization and method limitations. We appreciate your recognition and will make sure to incorporate the relevant clarifications and discussions in the final version of the paper.

---

### Official Review · Reviewer_sLTv · 2025-07-03

**Clarity:** 3
**Significance:** 3
**Originality:** 2
**Rating:** 4
**Confidence:** 4

**Summary:**

This paper proposes a multimodal tabular reasoning framework called TURBO: it utilizes structured table content as input to DeepSeek to generate reasoning paths corresponding to the given questions. After further improving the data quality, SFT and GRPO post-training techniques are employed to train multimodal visual understanding models, enhancing their reasoning capabilities. Based on Ovis2, results show that TURBO achieves significant improvements across multiple table understanding tasks, include TABMWP, WTQ, HiTab, TAT-QA, TabFact, InfoTabs, and MMMU.

**Questions:**

Please refer to the Strengths and Weaknesses.

**Ethical Concerns:**

["NO or VERY MINOR ethics concerns only"]

**Final Justification:**

As my concern is solved, I would like to raise my score from 3 to 4.

**Limitations:**

Please refer to the Strengths and Weaknesses.

**Paper Formatting Concerns:**

No formatting concerns.

**Quality:**

3

**Strengths And Weaknesses:**

Strengths:

1. The narrative of the article is clear. Overall, the proposed method corresponds well to the challenges highlighted in the paper. For example, it first introduces the limitations of multimodal table understanding, such as image information and content representation, and then proposes a two-stage training scheme—sending the table parsing results to DeepSeek to obtain QA pairs with reasoning paths. Subsequently, the visual modality is incorporated, and incremental training is conducted on VLM.

2. The experimental results in the "Main Results" section are relatively clear. TURBO demonstrates better metrics across multiple dimensions, not only in table understanding scenarios but also in validating general capabilities.


Weaknesses:

1. The originality is relatively weak. As mentioned in part 4, TURBO mainly proposes a complete data production and training pipeline for table understanding scenarios. For instance, reasoning data is generated using reasoning-capable LLMs, and post-training of VLM employs common methods like SFT and GRPO. These all seem to be relatively conventional methods.

2. In the main results (Table 1), most of the comparison methods for TURBO are non-reasoning models. From the perspective of the main experimental table, this comparison seems somewhat unfair. It would be better to include conclusions involving reasoning models.

3. Is the main innovation of this paper the two-stage framework? If so, the "Further Studies" section (5.3) seems to lack sufficient elaboration to highlight this aspect.

4. Is there a plan to contribute to community development? For example, will the related training data be further open-sourced?

---

> ### Author Rebuttal · Authors · 2025-07-31
>
> We appreciate your recognition of our writing, experimental results, and your valuable comments. We will answer the questions below, and we hope this clears up your concerns.
>
> **Q1:** The originality is relatively weak. Is the main innovation of this paper the two-stage framework?
>
> **A1:**  We agree that the individual components we employ, such as SFT and GRPO, are established and powerful techniques. However, we respectfully argue that the primary novelty of TURBO lies not in the invention of these base components, but in **the novel problem formulation and the synergistic framework we designed to address a challenging and practical task**: enhancing multimodal reasoning on table *images* by leveraging structured data as *privileged information* during training.
>
> Our key contributions to novelty are:
>
> 1. **A Novel Application of Learning with Privileged Information:** While the concept of privileged information exists, its application to bridge the significant modality gap between structured text (as the privileged information) and visual table images (as the inference-time input) for complex, multi-step reasoning is a novel contribution of our work. Prior works often require the structured data at inference time (e.g., HIPPO) or do not explicitly frame the problem this way. Our work directly tackles the real-world scenario where only table images are available for deployment, which is a practical and underexplored problem setting.
> 2. **The Modality-Bridging Data Generation Pipeline:** Our data generation process is more than a conventional use of LLMs. We introduce a **"structure-aware reasoning trace generator"** (line 12) specifically designed to create high-quality, modality-invariant reasoning paths. The key insight is that while the input modalities (structured table vs. table image) differ, the underlying logical steps to answer a question should be the same. By generating these explicit reasoning traces from the privileged information and using them to supervise the MLLM, we effectively create a "bridge" to transfer complex reasoning skills across the modality gap. The careful application of reject sampling (lines 208-212) further ensures the quality of this bridge, a crucial step for the success of the framework.
> 3. **A Synergistic and Effective Training Framework:** The originality also stems from the way our two-stage training process is tailored to this specific task.
>    - **Supervised Fine-Tuning (SFT)** on the bridged data is the foundational step that enables the MLLM to "internalize a 'think-then-answer' paradigm" (line 226) and learn the semantics of tabular reasoning from visual input.
>    - **Reinforcement Learning (GRPO)** then goes a step further, refining this nascent capability by allowing the model to explore a broader space of reasoning paths and reinforcing the most accurate and robust ones. As shown in our ablation study (Figure 4), both stages are complementary and crucial, leading to a significant average performance gain over the baseline.
>
> The effectiveness of this novel combination is demonstrated by our state-of-the-art results. TURBO achieves a **+7.2% average improvement over the previous SOTA with only 9k training examples**. This substantial gain would not be possible if our work were merely a conventional application of existing methods; rather, it is direct evidence of the effectiveness of our proposed framework.
>
> We hope this clarifies the originality and contribution of our work. We will ensure these points are further emphasized in the final version of the paper.
>
> **Q2:** It would be better to include conclusions involving reasoning models.
>
> We conducted **additional experiments** to compare our 8B TURBO model against leading large-scale, reasoning-focused models, even those that operate on structured text inputs, which is an inherently easier task. We compared TURBO with **DeepSeek-R1 (671B)**, a state-of-the-art reasoning LLM, and **QvQ (72B)**, another powerful model designed by Qwen team known for its reasoning capabilities. The results are summarized below:
>
> | | TABMWP| WTQ| HiTab| TAT-QA | TabFact | InfoTabs  |
> |-|-|-|-|-|-|-|
> | QvQ-72B | 96.38  | 79.54  | 79.03  | 78.26  | 93.25   | 74.91|
> | DeepSeek-R1-671B | 98.31  | 85.03  | 82.86  | 86.79  | 91.61   | 79.46|
> | Turbo-8B | 96.75  | 67.80  | 72.15  | 73.21  | 85.81   | 81.89 |
>
> This comparison leads to several important conclusions:
>
> 1. **Remarkable Performance Despite Modality Disadvantage:** TURBO operates on raw table images, which requires it to first solve the complex perception problem of parsing the table structure and content before it can even begin to reason. In contrast, models like DeepSeek-R1 are given clean, structured Markdown tables—the very privileged information we use only during training. Despite this significant disadvantage in input modality, our 8B model achieves performance that is highly competitive with a 671B model, and even **surpasses DeepSeek-R1 on the InfoTabs dataset**. Our model also surpasses QvQ on TABMWP and InfoTabs datasets with smaller model size, which is a larger and stronger reasoning model.
> 2. **Exceptional Parameter Efficiency:** Our TURBO framework achieves these strong results with a model that is nearly **85 times smaller** than DeepSeek-R1 (8B vs. 671B). This demonstrates that our proposed training methodology is highly effective at distilling complex reasoning abilities into a much more efficient model. The strength of TURBO is not in model scale, but in its novel technique for bridging the modality gap and transferring reasoning skills.
> 3. **Validation of our Core Claim:** This comparison strongly validates our central thesis: by leveraging privileged information during training, we can equip a moderately-sized MLLM with reasoning capabilities on visual tables that approach, and in some cases exceed, those of massive LLMs working with clean, structured data.
>
>
> **Q3:** Is there a plan to contribute to community development? For example, will the related training data be further open-sourced?
>
> **A3:** We promise we will release the related training data and model checkpoint for community development.

---

> > ### Comment · Reviewer_sLTv · 2025-08-05
> >
> > Thank you for your detailed explanation regarding my concern. From my perspective, the further emphasis in Q1 may be beneficial in the final submission. As my concerns is solved, I will raise my score to 4.

---

> > > ### Author Response · Authors · 2025-08-06
> > >
> > > Thank you very much for your feedback and for raising your score. We’re glad that our response was able to address your concern. We will make sure to revise the paper accordingly in the final version regarding Q1.

---

### Decision · Program_Chairs · 2025-09-17

**Decision:**

Accept (poster)

**Comment:**

The paper introduces TURBO, a framework for multimodal tabular reasoning that leverages privileged structured information during training to improve reasoning over table images. By combining a structure-aware reasoning trace generator, supervised fine-tuning, and reinforcement learning, TURBO achieves strong performance across multiple benchmarks and demonstrates high parameter efficiency. While reviewers initially raised concerns about originality, baseline comparisons, and limitations, the rebuttal addressed these thoroughly with new experiments, clearer motivation, and detailed failure analyses. Given its solid methodology, scalability, and state-of-the-art results, I recommend acceptance.